# POEMS Syndrome: Presented as Idiopathic Multicentric Castleman Disease of Plasma Cell Variant for Eight Years and Dramatic Treatment with Siltuximab Followed by Autologous Peripheral Blood Stem Cell Transplantation

**DOI:** 10.3390/diagnostics12040998

**Published:** 2022-04-15

**Authors:** Yong-Moon Lee, Yoon Seok Choi, Jin-Man Kim

**Affiliations:** 1Department of Pathology, College of Medicine, Dankook University, Cheonan 31116, Korea; vilimoon@dkuh.co.kr; 2Department of Hematology-Oncology, College of Medicine, Ajou University, Suwon 16499, Korea; wyfran@ajou.ac.kr; 3Department of Pathology, College of Medicine, Chungnam National University, 266 Munwha-ro, Jung-gu, Daejeon 35015, Korea

**Keywords:** POEMS, iMCD, siltuximab

## Abstract

Background: POEMS syndrome (POEMS) is a rare plasma cell clonal paraneoplastic syndrome consisting of polyneuropathy, organomegaly, endocrinopathy, monoclonal protein, and skin changes presenting with idiopathic multicentric Castleman disease (iMCD) histology, the treatment of which has not yet been well established. iMCD is also a distinctive rare non-clonal lymphoproliferative disorder, of which dramatic response to Siltuximab, a monoclonal anti-IL-6 antibody, has been reported recently. Methods: the differential diagnosis between POEMS and iMCD can be very challenging because of the identical histology, overlapping similar symptoms such as polyneuropathy, and vital signs insidiously presented to diagnose POEMS. Results: here, we report the case of a 53-year-old man with iMCD treated for 8 years developing sequential polyneuropathy, monoclonal gammopathy, and bone lytic lesions, all of which were confirmed after his iMCD achieved complete remission resulting from siltuximab administration and finally confirmed as POEMS. Conclusions: we describe the clinical ambiguity of disease presenting that we can face in the real world between iMCD and POEMS and emphasise careful, enduring observation lasting several years.

## 1. Introduction

POEMS syndrome (POEMS) is a rare paraneoplastic syndrome consisting of polyneuropathy, organomegaly, endocrinopathy, monoclonal protein, and skin changes in plasma cell neoplasms [1]. It was initially reported in Japan [2,3], and a worldwide survey reported a prevalence of approximately 0.3 per 100,000 [4]. However, diagnosis is very challenging due to the insidious onset of the variability of clinical manifestations. Idiopathic multicentric Castleman disease (iMCD) is a distinctive rare non-clonal lymphoproliferative disorder involving multiple sites characterised by distinct follicles with expanded mantle zones of small lymphocytes forming concentric rings surrounding germinal centres; its aetiology is unknown [5,6]. Not only does iMCD infrequently show peripheral neuropathy, but 11–30% of POEMS patients have documented clonal plasma cell disease with iMCD-like histology, which complicates the diagnosis of POEMS in the practical era [7]. Siltuximab, a human-mouse chimeric immunoglobulin G1k monoclonal antibody against human IL-6(Interleukin-6), is a promising therapeutic option for MCD [8,9]. Herein, we describe the successful therapeutic rescue of POEMS presenting with exceptionally 8-year lasting iMCD, who suffered disease progression despite full cycled CHOP and MINE chemotherapies and supportive care, with siltuximab followed by autologous peripheral blood stem cell transplantation (PBSCT).

## 2. Case Report

A 53-year-old man with tan-coloured skin (Figure 1a) complained of right inguinal discomfort caused by a large enlarged lymph node. He denied any systemic symptoms such as fever, malaise, and weight loss, and his serologic test was unremarkable. The abdominal CT scan detected multiple iliopelvic lymph nodes with variable sizes. He had idiopathic MCD (iMCD) of unknown aetiology for 8 years diagnosed by us before. When he visited us again, multiple enlarged lymph node chains were noted along the right iliopelvic region, approximately 7 cm on CT scan (Figure 1b,c), and serologic evaluations were unremarkable, including serum vascular endothelial growth factor (VEGF). He was a human immunodeficiency virus (HIV)-and human herpes Virus-8 (HHV8)-free immunocompetent person, leading to the diagnosis of iMCD. According to the iMCD regimen, systemic prednisolone followed by cyclophosphamide, adriamycin, vincristine, and prednisolone (CHOP) chemotherapy was administered.

Nevertheless, disease progression was observed; therefore, mesna, ifosfamide, mitoxantrone, etoposide (MINE), the second-line chemotherapy, was administered. However, no changes were identified, no further treatments were determined, and he had been under close observation until then. An inguinal lymph node excisional biopsy was again performed to investigate recent disease progression, which was conglomerated and encapsulated (Figure 1d). Microscopic examination revealed many prominent but atrophic germinal centres with hyalinisation (Figure 1e). The concentric mantle zone arranged lymphocytes and abundant plasma cells were widely distributed in the interfollicular area, and both CD138 and monotypic lambda light chain were positive in the immunohistochemical study (Figure 1f), which was identical to his initial diagnosis (Figure 1g). No M protein restriction was proven in the serum electrophoresis examination, so the diagnosis of iMCD with monotypic lambda light-chain plasma cells was still reasonable. The only concern was the nuclear atypism of monotypic lambda light-chain plasma cells (Figure 1h–l) presented since his initial lymph node evaluation, suspicious for plasma cell neoplasm histologically but failed to prove it by other methods. Because there were no responses to previous cytotoxic chemotherapies, siltuximab, a new noticeable therapy for iMCD, was administered to rescue disease progression. After the treatment, his inguinal and iliopelvic lymph nodes were not detected on physical examination and CT scan dramatically at all (Figure 1o,p), which was accompanied by less-coloured skin in his trunk, an unexpected outcome (Figure 1m). However, sudden numbness and weakness occurred in the right lower limb. Unfortunately, these symptoms gradually spread from the feet to the knees and finally developed into gait disturbance with lytic lumbar lesions (Figure 1n). Subsequent electromyography demonstrated a demyelinating polyneuropathy with secondary axonal degeneration making a severe consideration of POEMS. POEMS is a rare paraneoplastic syndrome consisting of polyneuropathy, organomegaly, endocrinopathy, monoclonal proteins, and skin changes that arise in plasma cell neoplasms [1]. It was initially reported in Japan [2,3], and a worldwide survey reported a prevalence of approximately 0.3 per 100,000 [4]. However, diagnosis is very challenging due to the insidious onset of the variability of clinical manifestations. Idiopathic multicentric Castleman disease (iMCD) is a distinctive rare non-clonal lymphoproliferative disorder involving multiple sites, characterised by distinct follicles with expanded mantle zones of small lymphocytes forming concentric rings surrounding germinal centres, the aetiology of which is unknown [5,6]. Not only does iMCD infrequently show peripheral neuropathy, but 11–30% of POEMS patients have a documented clonal plasma cell disease with iMCD-like histology, which complicates the diagnosis of POEMS in the practical era [7]. Siltuximab, a human–mouse chimeric immunoglobulin G1k monoclonal antibody against human IL-6, is a promising therapeutic option for MCD [8,9]. Herein, we describe the successful therapeutic rescue of POEMS presenting with exceptionally 8-year lasting iMCD with siltuximab followed by autologous peripheral blood stem cell transplantation (PBSCT).

Serum electrophoresis revealed a monoclonal gammopathy composed of IgA/lambda protein, which was not detected before, and lytic bone lesions were noted in the lumbar and pelvic bones (Figure 1). Interestingly, the serum vascular endothelial growth factor (VEGF) levels were consistently within the normal range. Having telltale symptoms and signs leading to the diagnosis of POEMS and confirming complete remission (CR) status due to the dramatic effect of siltuximab, we switched the regimens to melphalan followed by PBSCT a POEMS regimen, hoping for disease-free status (Figure 1).

## 3. Discussion

POEMS syndrome is a rare and multiple-organ systems disorder involving paraneoplastic syndrome of plasma cell neoplasm presenting with polyneuropathy, organomegaly, endocrinopathy, monoclonal protein almost lambda light-chain-type with skin changes [1]. Polyneuropathy, often a dominant clinical feature of POEMS, presents subacutely in a length-dependent manner with distal symmetric sensory symptoms, such as tingling burning followed by weakness. Patients often have significant lower-limb involvement with bilateral foot drop and atrophy. Nerve conduction studies and electromyography help demonstrate the primary demyelinating process with secondary axonal injury [10]. Between 11% and 30% of POEMS patients show MCD-like histology, one of the major diagnostic criteria. The detailed diagnostic criteria are listed in Table 1. Castleman disease (CD), another distinct non-clonal lymphoproliferative disorder that Benjamin Castleman first described in 1950, is worth discussing [11]. CD shows a typical expanded mantle zone of small lymphocytes forming concentric rings surrounding germinal centres, which can be further classified depending on whether prominent hyalinised vascularity is demonstrated (hyaline vascular variant) or abundant plasma cells are demonstrated (plasma cell variant), whether involved legion-localized (unicentric CD) or MCD. The development of MCD has shown a strong correlation with HIV and HHV-8 infections, which is thought to be the aetiology of MCD. The rest of the MCD cases failed to demonstrate that HIV and HHV-8 infections were classified as iMCD [9]. Polyneuropathy occurs in approximately 27% of the patients with CD. It typically presents as distal lower limb numbness. Approximately half of the patients reported positive sensory symptoms such as tingling or burning or motor symptoms such as weakness. Pain is often absent. Neurological examination demonstrates length-dependent sensory deficits in the lower limbs, rarely accompanied by real weakness. The polyneuropathy of CD is similar to that of POEMS with motor and sensory polyneuropathy or polyradiculoneuropathy but is less severe. Making the differential diagnosis between POEMS and iMCD can be very challenging due to the identical histology, overlapping similar symptoms such as polyneuropathy, and vital signs insidiously presented to diagnose POEMS [10]; only those with polyneuropathy and plasma cell clonality mostly showing lambda light-chain restriction should be classified as classic POEMS [1]. Patients can be classified as having CD variants of POEMS without both of these characteristics. Despite the relationship between disease activity and serum VEGF levels, successful outcomes have been associated with directing therapy to the underlying clonal CD rather than only targeting VEGF with anti-VEGF antibodies [1]. Owing to the paucity of published randomised clinical trials on POEMS, treatment recommendations are based on limited trial data and case series. Some case series have shown a favourable response to melphalan with PBSCT [1]. In comparison to POEMS, iMCD involves multicentric lymphadenopathy and systemic inflammation, such as peripheral neuropathy, all of which result from a cytokine storm, including IL-6. In symptomatic iMCD, high-dose steroids, rituximab, or a combination of conventional chemotherapy regimens (CHOP) are used [8]. A recently published real-world experience showed a dramatic response to siltuximab, a monoclonal antibody against IL-6. Interestingly, VEGF and IL-6 are closely correlated; the latter has been shown to stimulate VEGF production in monoclonal plasma cells in the previous study [1,9]. Our patient showed iMCD histology without any cryptic signs of POEMS, such as monoclonal M-protein in serum electrophoresis, serum VEGF elevation, polyneuropathy, or sclerotic bone lesions, for 8 years. The nuclear atypism of lambda light-chain-restricted monotypic plasma cells was not comprehended as an iMCD. However, beginning with his gait disturbance, all signs under the diagnostic criteria of POEMS mentioned above appeared abruptly in the course of siltuximab administration, which showed a dramatic effect on his clonal iMCD component. After confirming the absence of detectable lymph node chains in the inguinal and iliopelvic areas on the CT scan, we switched the regime to melphalan with PBSCT, focusing on POEMS for consolidation, and CR was achieved.

We encountered a unique patient with final POEMS who presented with iMCD composed of monotypic lambda light-chain-restricted plasma cells for 8 years who showed CR after siltuximab administration followed by melphalan with PBSCT. No clinical evidence for POEMS was identified, amazingly, for 8 years, except for nuclear atypism of monotypic lambda light-chain-restricted plasma cells consisting of iMCD. Both diseases can progress insidiously, and the clinical features of one disease include all the clinical features of the opposite disease; therefore, when our conclusion that the shape of plasma cells may be atypical microscopically in POEMS was accepted, at last, it is never easy to differentiate them. For example, a 60-year-old lady who suffered from both diseases for 30 years was reported in [12]. Therefore, it can be concluded that the histologic features of plasma cells could be an early detectable point in predicting iMCD clinical behaviour, and siltuximab can be considered as a good treatment option for not only iMCD but also POEMS presenting as iMCD clinically and pathologically, which has not been persuaded before. These findings might provide a clue to elaborating the pathogenesis of POEMS and iMCD, and further studies, including targeted genomic sequencing, are needed to predict the clinical outcomes and discriminate these two overlapping diseases properly.

Despite our long-standing care and support, our patient failed to survive because of his poor general condition, leading to multiple organ failure.

## 4. Conclusions

Since the symptoms of POEMS are gradual and the histological findings are the same as those of iMCD, it is practically impossible to differentiate these two diseases completely. For this reason, the moment of missing the golden opportunity to accurately prescribe IL-6 agents by diagnosing POEMS at an early stage. Although not included in the diagnostic criteria of POEMS, we suggest the possibility of diagnosing as POEMS if morphological abnormalities of plasma cells are continuously observed microscopically through this unique experience.

## Figures and Tables

**Figure 1 diagnostics-12-00998-f001:**
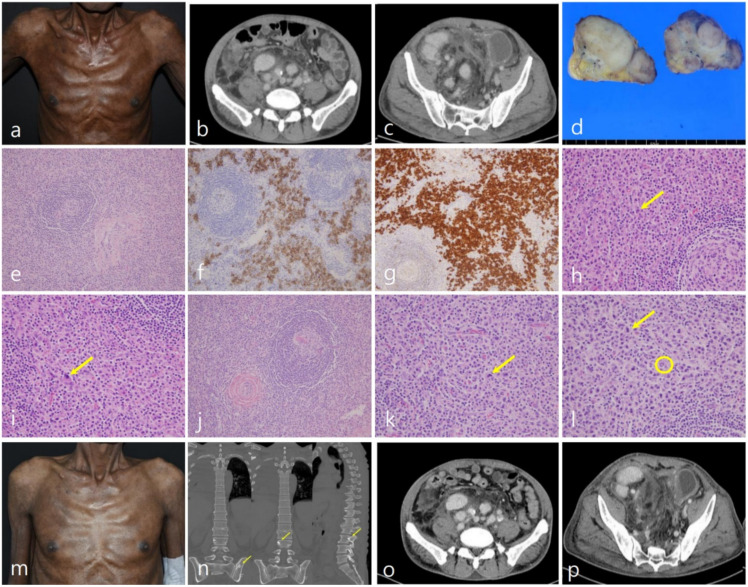
Tan-coloured skin developed gradually (**a**). Enlarged lymph node chains in right inguinal (**b**) and iliopelvic area (**c**) approximately 7 cm in CT scan. An excised inguinal lymph node showing vague fusion but complete encapsulation (**d**). Relatively atrophied germinal centre surrounded by concentrically arranged mantle zone with hyalinized vessels (**e**; H&E stain, ×100), which consists of both CD138 (**f**; CD138 IHC stain, ×200) and lambda light chain strongly positive (**g**; lambda IHC stain, ×200) plasma cells. Unignorable nuclear atypism of plasma cells (**h**,**i**; yellow arrow, H&E stain, ×400) making the neoplastic condition suspicious. Initial microscopic findings identical to those 8 years later, as described earlier (**j**; H&E stain, ×100) also demonstrating atypical plasma cells (**k**; yellow arrow, H&E stain, ×400) having the Dutcher body (**l**, yellow circle, H&E stain, ×400) Less-coloured skin (**m**), bone sclerotic lesions (**n**; yellow arrow), and dramatically disappeared lymph node chains (**o**,**p**) after Siltuximab administration.

**Table 1 diagnostics-12-00998-t001:** Criteria for the diagnosis of POEMS syndrome [1].

Mandatory Major Criteria	Other Major Criteria (One Required)	Minor Criteria
Demyelinating polyneuropathy Monoclonal plasma cell proliferative disorder (almost always lambda)	Castleman disease Sclerotic bone lesions VEGF elevation	Organomegaly Extravascular volume overload Endocrinopathy Skin changes

POEMS syndrome was confirmed when both mandatory major criteria, one of the three other major criteria, and one of the six minor criteria were present.

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
