# Peer review of "POEMS Syndrome: Presented as Idiopathic Multicentric Castleman Disease of Plasma Cell Variant for Eight Years and Dramatic Treatment with Siltuximab Followed by Autologous Peripheral Blood Stem Cell Transplantation"

_diagnostics, 2022, doi:10.3390/diagnostics12040998_

Round 1
Reviewer 1 Report
Thank you for the opportunity for review. I would recommend adding some information about patient's symptoms and work up when first diagnosis of iMCD was made as mentioned in my former review. The other edits are appropriate.
Author Response
Thank you for your insightful recommendation. We missed some story flow, as your advice, we added our patient's original description led us to iMCD first, shortly as follows;
A 53-year-old man with tan-coloured skin (Fig) complained of right inguinal discomfort caused by a large enlarged lymph node. He denied any systemic symptoms such as fever, malaise, and weight loss, and his serologic test was unremarkable. The abdominal CT scan detected multiple iliopelvic lymph nodes with variable sizes
Reviewer 2 Report
While this article is interesting, it is known that MCD
- Please reword line 43, 44 to describe the sequential changes. In its current form, it is not very clear on the application of siltuximab.
- In fig1., please mark the figures, it is difficult to follow.
- Please provide a higher resolution for the H&E images, it is not clear.
- The CT scans should have similar sectional imaging to compare the past history with the current one.
- VEGF levels being normal seems to be inconsistent with POEMS (ref: Vascular endothelial growth factor as a predictive marker for POEMS syndrome treatment response: retrospective cohort study)
- Was papilloedema detected? Typically disc swelling may be observed although it is confounded by normal VEGF.
Author Response
Thank you for the insightful recommendation, which will make our work more complete, I am sure!. As your advice, we supplemented the descriptions as follows;
1. Herein, we describe the successful therapeutic rescue of POEMS presenting with exceptionally eight-year lasting iMCD, who suffered disease progression despite full cycled CHOP and MINE chemotherapies and supportive care, with siltuximab followed by autologous peripheral blood stem cell transplantation (PBSCT).
2. We attached the fig1 file and word file marked as you can see. We sincerely apologize to make it difficult to read.
3. To be honest, we performed three inguinal lymph node excisional biopsies to
discover why our patient suffered disease progression despite the intensive treatments we gave and to understand the atypical plasma cells in iMCD lasting eight years! Would you mind choosing the best image for me? All we have are attached in the linked file in my cloud.
https://dkuniv-my.sharepoint.com/:b:/g/personal/12200301_dankook_ac_kr/Ea6FRXZxQHtIpbuXH5g84EMBg6K8gp-YzsteCa4L0dRD4w?e=NVYrtC
4. Would you mind choosing the best image for me? All we have are attached in the linked file in my cloud.
https://dkuniv-my.sharepoint.com/:b:/g/personal/12200301_dankook_ac_kr/Ea6FRXZxQHtIpbuXH5g84EMBg6K8gp-YzsteCa4L0dRD4w?e=NVYrtC
5. The initial level of serum VEGF was normal, but when neurologic symptoms were developed in our patient, serum VEGF was 7,292 pg/mL
6. Unfortunately, papilloedema was not found.

Round 2
Reviewer 2 Report
Thank you for the changes.
Please use slide 36/37 to denote nuclear atypia in plasma cells.
Please consider using slide 11 for the post siltuximab treatment CT.
Author Response
Dear reviewer#2
I appreciate your insightful recommendation and help in choosing the best images, strengthening our conclusion. We revised the figure as per your instruction.
Slides 36/37 have been inserted as h, and I boxes, and slide 11 has been inserted as b and c boxes before Siltuximab treatment, as o and p boxes after Siltuximab treatment, separately. We hope our correlations will make you satisfied and be suitable for publication. Thank you.
With regards

This manuscript is a resubmission of an earlier submission. The following is a list of the peer review reports and author responses from that submission.
Round 1
Reviewer 1 Report
The authors presented a patient with POEMS syndrome who initially diagnosed with iMCD. Since the symptoms are similar between MCD and POEMS syndrome, differentiating between MCD and POEMS syndrome may be difficult. This is a single case report, not a review article. The structure should contain introduction, case report, and discussion. The authors may describe MCD and POEMS syndrome and the similarity of the diseases in Introduction, then a case report in Case Report, and discuss difficulty of differential diagnosis of MCD and POEMS syndrome and the efficacy of treatment with IL-6 blockade for POEMS syndrome in Discussion. If the authors think iMCD turned into POEMS syndrome, write so. Patient with MCD and POEMS syndrome was already reported about 30 years ago (Scand J Rheumatol. 1994;23(4):215-7). The authors should mention the case report.
Author Response
Dear repected reviwer
Q: The authors presented a patient with POEMS syndrome who initially diagnosed with iMCD. Since the symptoms are similar between MCD and POEMS syndrome, differentiating between MCD and POEMS syndrome may be difficult. This is a single case report, not a review article. The structure should contain introduction, case report, and discussion. The authors may describe MCD and POEMS syndrome and the similarity of the diseases in Introduction, then a case report in Case Report, and discuss difficulty of differential diagnosis of MCD and POEMS syndrome and the efficacy of treatment with IL-6 blockade for POEMS syndrome in Discussion.
A: Thank you so much for reading my article and giving me your heartfelt advice. As you said, it has been modified according to the case report format. Differentiation between the two diseases is very challenging, but I am sharing my experiences as a pathologist, hoping that the strange shape of plasma cells will be helpful.
Q: If the authors think iMCD turned into POEMS syndrome, write so. Patient with MCD and POEMS syndrome was already reported about 30 years ago (Scand J Rheumatol. 1994;23(4):215-7). The authors should mention the case report.
A: I have mentioned in my article the scarce case report you mentioned. Thank you.
Reviewer 2 Report
This is an interesting case but the manuscript needs significant edits.
The language used in describing the case is casual, at times appears incongruent with the rest of the text and often grammatically incorrect. Also, in the starting of case presentation, iMCD is made but the evidence leading to this diagnosis such as pathology and serology is not mentioned until later in the case when it is compared to the repeat testing. This impedes the flow of clinical description and leaves the reader guessing rather than forthright presenting the data for one to understand. Also, chemotherapy abbreviation such as MINE or PBSCT without prior description have been used in the manuscript. The Journal caters to a wider medical audience and not exclusively Hem-oncologists. It would be prudent to describe the regimen before the abbreviations are used. The authors need to elaborate more in the clinical description of the case about the difference in the evidence on two presentation including the electrophoresis before the correlation between POEMS and MCD is presented to the readers.
In the discussion section, there is some redundancy with the introduction section. Also, some of the text seem to be incoherent specifically line 154, which does not seem to confer any meaning what so ever.
Overall, this manuscript needs considerable editing before it can be considered for publication.
Author Response
Dear respected reviewer
Q:
This is an interesting case but the manuscript needs significant edits.
The language used in describing the case is casual, at times appears incongruent with the rest of the text and often grammatically incorrect. Also, in the starting of case presentation, iMCD is made but the evidence leading to this diagnosis such as pathology and serology is not mentioned until later in the case when it is compared to the repeat testing. This impedes the flow of clinical description and leaves the reader guessing rather than forthright presenting the data for one to understand. Also, chemotherapy abbreviation such as MINE or PBSCT without prior description have been used in the manuscript. The Journal caters to a wider medical audience and not exclusively Hem-oncologists. It would be prudent to describe the regimen before the abbreviations are used. The authors need to elaborate more in the clinical description of the case about the difference in the evidence on two presentation including the electrophoresis before the correlation between POEMS and MCD is presented to the readers.
In the discussion section, there is some redundancy with the introduction section. Also, some of the text seem to be incoherent specifically line 154, which does not seem to confer any meaning what so ever.
Overall, this manuscript needs considerable editing before it can be considered for publication.
A: Thank you so much for reading my article and giving me your heartfelt advice. As you well know, it is challenging to differentiate between the two diseases, clinically and pathologically, almost impossible. As a pathologist, I am only speaking from my experience of the claim that the strange shape of plasma cells can be a clue to the fate of the two diseases. These conclusions were made more convincing after experiencing the effect of a surprising new drug called anti IL-6 agent. I sincerely apologize for the lack of readability in my writing, with some overlapping expressions and abbreviations. We have commissioned a professional English proofreading company and conducted intensive proofreading. We would be very grateful if you could review it again.
Round 2
Reviewer 2 Report
Thank you for the appropriate edits. However, I would recommend following changes to the manuscript.
1) The manuscript will benefit from a brief and succinct conclusion section which is missing in the current version. The "Patent" section is not required and patient outcome can be summarized in the case history.
2) In the case report section, it is unclear when the authors write "He had idiopathic MCD (iMCD) of unknown aetiology for eight years. When he first visited us...", if the patient was diagnosed elsewhere or by the authors. If he was diagnosed elsewhere and presented to the authors 8 years after the diagnosis, then it would be reasonable to mention the initial presentation and how the diagnosis was made.